# New Genetic Variants in *CYP2B6* and *SLC6A* Support the Role of Oxidative Stress in Familial Ménière’s Disease

**DOI:** 10.3390/genes13060998

**Published:** 2022-06-01

**Authors:** Sini Skarp, Johanna Korvala, Jouko Kotimäki, Martti Sorri, Minna Männikkö, Elina Hietikko

**Affiliations:** 1Research Unit of Biomedicine, Faculty of Medicine, University of Oulu, 90220 Oulu, Finland; sini.skarp@oulu.fi; 2University of Oulu Graduate School, University of Oulu, 90220 Oulu, Finland; johanna.korvala@oulu.fi; 3Department of Otorhinolaryngology, Kainuu Central Hospital, 87300 Kajaani, Finland; jouko.kotimaki@kainuu.fi; 4Department of Otorhinolaryngology and Head and Neck Surgery, Oulu University Hospital, 90220 Oulu, Finland; martti.sorri@oulu.fi; 5Finland & PEDEGO Research Unit, University of Oulu, 90220 Oulu, Finland; 6Infrastructure for Population Studies, Northern Finland Birth Cohorts, Faculty of Medicine, University of Oulu, 90220 Oulu, Finland; minna.ruddock@oulu.fi; 7Center for Life-Course Health Research, Faculty of Medicine, University of Oulu, Finland and Oulu City Hospital, 90220 Oulu, Finland

**Keywords:** Ménière’s disease, genetics, whole exome sequencing

## Abstract

The objective was to study the genetic etiology of Ménière’s disease (MD) using next-generation sequencing in three families with three cases of MD. Whole exome sequencing was used to identify rare genetic variants co-segregating with MD in Finnish families. In silico estimations and population databases were used to estimate the frequency and pathogenicity of the variants. Variants were validated and genotyped from additional family members using capillary sequencing. A geneMANIA analysis was conducted to investigate the functional pathways and protein interactions of candidate genes. Seven rare variants were identified to co-segregate with MD in the three families: one variant in the *CYP2B6* gene in family I, one variant in *GUSB* and *EPB42* in family II, and one variant in each of the *SLC6A*, *ASPM*, *KNTC1*, and *OVCH1* genes in family III. Four of these genes were linked to the same co-expression network with previous familial MD candidate genes. Dysfunction of *CYP2B6* and *SLC6A* could predispose to MD via the oxidative stress pathway. Identification of *ASPM* and *KNTC1* as candidate genes for MD suggests dysregulation of mitotic spindle formation in familial MD. The genetic etiology of familial MD is heterogenic. Our findings suggest a role for genes acting on oxidative stress and mitotic spindle formation in MD but also highlight the genetic complexity of MD.

## 1. Introduction

Ménière’s disease (MD) is an inner ear disorder described by Prosper Ménière in 1861. It is characterized by tinnitus, sensorineural hearing impairment, and spells of spinning vertigo. Endolymphatic hydrops (EH), a fluid overload in the endolymphatic system, has been associated with MD since it was observed in patients by two independent temporal bone histological studies in 1938 [1,2]. However, EH has also been observed in subjects without MD and in the contralateral asymptomatic ear of MD patients, suggesting that EH is not the sole cause of the symptoms in MD [3]. As the sensitive structures of the inner ear cannot survive in vivo exploration and sufficient imagining techniques have only recently started to develop, MD has proven a difficult disease to study

The differences in the population and race incidences of MD have provided evidence of the genetic predisposition of MD [4,5]. Cases of familial clustering have been reported worldwide and familial MD has been reported in 5–23.5% of all cases [6]. Most Ménière families show autosomal dominant inheritance with reduced penetrance [7]. 

Molecular genetic study of MD is complicated by genetic and clinical heterogeneity and diagnostic challenges [6]. A locus on chromosome 12p12.3 has been reported for Swedish familial MD [8]; however, this was not replicated in Finnish families [9] and candidate gene studies have not been published from this locus. However, next-generation sequencing techniques have provided new information: rare variants segregating with familial MD in a family with the sequence similarity 136 member A (*FAM136A)*, dystrobrevin α *(DTNA)*, protein kinase C β *(PRKCB)*, dermatopontin (*DPT)*, and semaphorin 3D *(SEMA3D*) genes have been identified by exome sequencing [10,11,12]. So far, the identified variants have been unique to the studied families. Exome sequencing has also revealed a build-up of rare variants in genes expressed in the inner ear in a case of childhood-onset MD, suggesting a multifactorial genetic etiology [13]. 

Gallego-Martinez and colleagues (2019) studied sporadic MD patients and found a significant enrichment of missense variants in genes associated with hearing loss, which was not observed in the controls [7]. Oh et al. (2019) observed 15 rare heterozygous variants in previously MD-associated candidate genes in 12 sporadic patients [14]. Studies utilizing a targeted sequencing panel have reported a burden of missense variants in several genes, including the NADPH oxidase 3 *(NOX3)* and netrin 4 (*NTN4)* genes participating in axonal guidance signaling and otogelin (*OTOG)* encoding the extracellular matrix (ECM) component of the inner ear [15,16]. Variants in *OTOG* and otogelin like (*OTOGL)*, otopetrin 1 (*OTOP1)*, tectorin α (*TECTA)*, zinc finger protein 91 (*ZNF91)*, and armadillo repeat containing 9 (*ARMC9)* were associated with vertigo in a recent genome-wide meta-analysis [17]. Overall, MD appears to have a complex genetic background affected by multiple genes and pathways.

The role of genetic factors in the development of MD has been established. Although the genetic study of MD is challenging, genetics provides a safe way of studying the aetiology of MD. The growing amount of information on the genetic and biological processes behind MD can help to discover the disease mechanism in the future. Here, we present the results of exome sequencing of three Finnish families with familial MD.

## 2. Materials and Methods

### 2.1. Families

Three MD families, each with three patients with diagnosed MD according to the American Academy of Otolaryngology-Head and Neck Surgery (AAO-HNS) 1995 criteria, were recruited to the study [18]. Patient records of the participants were examined, and all patients were phone interviewed. If acceptable diagnostic studies had not been conducted (clinical examination, audiogram, magnetic resonance study), the subject was referred to a specialist in otology. Family pedigrees are presented in Figure 1. Families have been described in more detail previously [6,9]. Clinical data was updated before genetic analyses. The status of patient IV:1 in family 1 was updated from possible to definite MD. In the genetic analyses, whole exome data of 76 unrelated individuals was used as a control dataset. These individuals have the same ethnicity as MD families and are from the same geographical region in northern Finland. Hospital records confirmed that none of the controls had MD or had been studied for Ménière-like symptoms. This research was approved by the Ethics Committee of the Northern Ostrobothnia Hospital District (20.12.2000, approval number 131/2000). Subjects were informed of the risks involved in obtaining a venous blood sample and gave their written consent to participate in the study.

### 2.2. Whole Exome Sequencing

Venous blood samples were collected from all subjects. Genomic DNA was extracted from 3 mL of EDTA blood using an Ultraclean Blood DNA Isolation Kit (Mo Bio Laboratories, Carlsbad, CA, USA). The quality of DNA was measured using a nano drop to determine the concentration and purity of samples. Samples were subjected to whole exome sequencing. The whole exome data was acquired from BGI Tech Solutions, Hong Kong. They performed exome capture, alignment, and variant calling. A SureSelect Agilent V4 (51M) kit (Agilent Technologies, Santa Clara, CA, USA) was used for exome capture and the targets were sequenced with an IlluminaHigSeq2000 100PE platform (Appendix A). The reads were aligned to the hg19 human reference genome (Appendix A). A Genome Analysis ToolKit (GATK) was used for variant calling and filtering. The quality parameters used for variant filtering of single nucleotide polymorphisms (SNPs) were a root mean square of the mapping quality (MQ) < 40, Mapping Quality Rank Sum < −12.5, Quality by depth (QD) < 2, FisherStrand > 60, and Rank Sum Test for relative positioning of REF versus ALT alleles within reads (ReadPosRankSumTest) < −8. The variation filtering quality parameters for small insertions and deletions (indels) were FisherStrand >200, QD < 2.0, and ReadPosRankSumTest < −20.

### 2.3. Whole Exome Data Analysis

BEDTools version 2.26.0 was used for variant extraction and comparison between controls [19]. Variants shared by the MD patients in each family were compared to controls. After sample comparison, the identified variants were annotated using ANNOVAR (Annotate Variation) version 2016Nov04 [20] The purpose of functional annotation was to identify variants that are very rare, likely to alter the encoded protein, and estimated to be harmful. Only protein altering (nonsynonymous and missense variants) and splice site variants were accepted for further analysis. The in silico estimations of pathogenicity used were the Combined Annotation Dependent Depletion (CADD) phred [21] and ANNOVAR LJB, also known as the dbNSFP database for functional annotations (SIFT, PolyPhen2 HDIV, PolyPhen2 HVAR, LRT, MutationTaster, MutationAssessor, FATHMM, GERP++, PhyloP, and SiPhy). A variant was considered pathogenic when it had CADDpherd value > 15 and it was considered pathogenic by at least two of the LJB database estimations. Because familial MD is extremely rare, variations with a minor allele frequency (MAF) higher than 0.001 were excluded. The MAFs for the Finnish population were obtained from the Genome Aggregation Database (gnomAD) database (https://gnomad.broadinstitute.org/, accessed on 18 April 2022). Visualization of the variants in the exome sequencing alignments was performed using the Interactive Genomic Viewer (IGV, igv.org/app, accessed on 18 April 2022). The PubMed Gene, Uniprot, GeneCards Human gene database, and Online Mendelian Inheritance in Man databases were used to study the functions of the genes. GeneMANIA (genemania.org, accessed on 18 April 2022) was used for functional network analysis of candidate genes [22]. Variants were validated using capillary sequencing with an ABI3500xL Genetic Analyzer (Applied Biosystems, Waltham, MA, USA) and genotyped from additional family members to find alleles that co-segregate with the phenotype. Copy number variants were detected using cn.MOPS [23] and annotated using the database of genomic variants to identify common copy number variants [24]. A flowchart of the methods used in this study is presented in Figure 2.

## 3. Results

Altogether, seven rare variants were identified to co-segregate with MD in the three families: one variant in family I, two variants in family II, and four variants in family III (Table 1, Appendix A). Visualizations of each variant in one affected family member and one healthy carrier are presented in the Appendix A. Only deletions and insertions commonly observed in the general population were observed in the copy number variant data of the families.

A total of 58,891 SNPs were shared by the MD-affected family members in family I. After excluding variants observed in the control dataset, 1513 variants remained. After functional annotation, one missense variant (c.200C>T/p.T67M rs138264188) in the cytochrome P450 family 2 subfamily B member 6 (*CYP2B6)* gene was observed. Genotyping of additional family members confirmed segregation with MD (Table 2).

A total of 65,626 shared SNPs were observed in the whole exome data of the MD subjects in family II, of which 2123 variants remained after excluding variants observed in the control dataset. After functional annotation, a total of eight rare single nucleotide variants (SNVs) estimated to be pathogenic remained. After genotyping more family members, six of these variants were excluded (Table 2). One missense variant (c.323C>T/p.P108L) and one nonsense variant (c.1089G>A/p.W363X, rs201351228) were identified in the glucuronidase β (*GUSB*) and erythrocyte membrane protein band 4.2 (*EPB42*) genes, respectively.

A total of 68,789 SNPs were shared by the MD patients in family III. After excluding variants observed in the control dataset, 2572 variants remained. After functional annotation, a total of six rare SNVs estimated to be pathogenic were observed. Four variants remained after genotyping unaffected family members: one nonsense variant c.1316T>A/p.L439X in the ovochymase 1 (*OVCH1*) gene and three missense variants c.5242A>C/p.T1748P in kinetochore-associated 1 (*KNTC1*), c.5207A>G/p.Q1736R in assembly factor for spindle microtubules (*ASPM*), and c.322G>C/p.V108L (rs775035174) in solute carrier family 6 member 7 (*SLC6A7*).

A geneMANIA functional network analysis was conducted. The identified candidate genes together with the genes *FAM136A*, *DTNA*, *PRKCB*, *DPT*, and *SEMA3D* identified in the previous familial MD exome sequencing studies were included in the analysis [10,11,12]. *KNTC1*, *ASPM, SLC6A7*, and *GUSB* were linked to the same co-expression interaction network with previously identified candidate genes, suggesting that these genes’ interplay may contribute to the onset of MD (Figure 3).

## 4. Discussion

We identified seven genetic variants in three families segregating with familial MD. Four of these variants are located in genes linked to the same co-expression network with previous familial MD candidate genes. 

In family I, we identified a c.200C>T/p.T67M (rs138264188) variant in *CYP2B6*. The *CYP2B* gene encodes an enzyme belonging to the cytochrome P450 superfamily. Most of the genes in the CYP 1–4 families encode for proteins in eicosanoids metabolism and participate in cell respiration as a catalyst of oxidation reduction and elimination of reactive oxygen species (ROS) [25]. Mutations causing a deficiency in the cytochrome system have been described to cause congenital abnormalities with associated hearing loss [26]. Polymorphisms of oxidative stress genes, including *CYP1A1*, have previously been associated with presbyacusis [27]. ROS-induced mitochondrial damage leading to cochlear hair cell apoptosis has been suggested as a mechanism of cochlear damage in noise trauma, ototoxicity, and age-related cochlear degeneration [28,29]. Cytochromes are inducible to environmental stimuli, such as diet, and chemical inducers, drugs that would also fit well with the episodic nature of MD [25].

In family II, the c.323C>T/p.P108L (rs1268678201) missense variant in *GUSB* and the c.1089G>A/ p.W363X (rs201351228) nonsense variant in *EPB42* segregated with familial MD. The *GUSB* gene encodes a lysosomal hydrolase enzyme called β-glucuronidase that degrades glycosaminoglycans, which are carbohydrate chains of the extracellular matrix. Loss-of-function mutations in *GUSB* cause an autosomal recessive lysosomal storage disease called mucopolysaccharidosis type VII (MPS VII, https://www.omim.org/entry/253220) (accessed on 18 April 2022), which manifests as skeletal dysplasia, cognitive impairment, heart abnormalities, and hearing loss [30]. Evidence of a mouse model study suggested that the sensorineural hearing loss in MPS VII arises from alterations in the mass and stiffness of cochlear structures or impaired sensory cell function and indicated the possibility of a vestibular component [31]. The *GUSB*, therefore, has many qualities that make it an excellent candidate gene for familial MD. EPB42 is highly expressed in blood, where it is known to impact the erythrocyte shape and mutations in *EPB42* are known to cause autosomal recessive hereditary spherocytosis type 5 (https://www.omim.org/entry/612690) (accessed on 18 April 2022), characterized by hemolytic anemia. As total loss of the protein has no described impact on hearing ability or vestibular function, and *EPB42* was not linked to the functional interaction network, such as *GUSB*, *EPB42* seems an unlikely candidate gene for familial MD.

In family III, four rare variants in four genes were identified in exome sequencing. A missense variant c.322G>C/p.V108L (rs775035174) was identified in *SLC6A7*. This gene belongs to the solute carrier family, a large group of membrane transport proteins, which transport sugars, amino acids, vitamins, nucleotides, metals, inorganic ions, organic anions, oligopeptides, and drugs [32]. Mutations in the solute carrier *SLC26A4* encoding the pendrin protein cause prelingual deafness with hydrops of the inner ear cavities [33]. Other members of the solute carrier family (*SLC22A4, SLC26A5, SLC17A8, SLC12A2)* have also been associated with non-syndromic hearing loss (hereditaryhearingloss.org). *SLC4A1* protein expression levels have been observed to decline in an experimental endolymphatic hydrops model of MD; thus, *SLC4A1* is considered protective against MD [34]. These properties and previous associations in this gene family make *SLC6A7* a promising candidate gene for MD in family III. Oxidative stress could be one of the possible mechanisms of injury as in an experimental pig model, oxidative stress and inflammation were caused by disruption of the balance of SLC transporters [35].

Two heterozygous variants in genes affecting mitotic spindle formation were also observed in family III: a missense variant c.5207A>G/p.Q1736R in *ASPM* and c.5242A>C/p.T1748P (rs1272541364) in *KNTC1*. Mutations of the *ASPM* gene are known to cause autosomal recessive primary microcephaly, where some patients also have hereditary hearing loss [36]. The *KNTC1* gene is required for the assembly of the dynein-dynactin complex onto kinetochores (https://www.uniprot.org/uniprot/P50748) (accessed on 18 April 2022). The dynein regulatory complex is known to be essential for ciliary motility and otolith biogenesis in the inner ear [37]. No links between the *OVCH1* gene and hearing or vestibular function were found in the literature and *OVCH1* was not linked to the functional interaction network and thus seems an unlikely candidate gene for familial MD.

In the functional interaction network, many of the genes are co-expressed, which indicates a potential for interaction. The network analysis shows that four of the identified candidate genes can be linked with genes identified by others, suggesting that although individual predisposing variants may be family specific, same molecular pathways are affected. The functions of the auditory and vestibular systems in MD are still at large partly unknown and a better understanding of the molecular mechanisms of the familial MD will eventually improve the understanding of MD etiology. Here, we report seven novel variants in as many candidate genes that may predispose to familial MD. Our present findings support the role of oxidative stress but also suggest dysregulation of mitotic spindle formation, highlighting the genetic complexity behind MD. 

This study has several limitations. Familial MD has been observed to have decreased penetrance and the families also have individuals with Ménière-like symptoms but without the full triad of MD [6]. This makes the genetic study of familial MD challenging and ruling out candidate genes based on the genetic status of healthy relatives is a possible limitation. As MD typically manifests in middle age, it is challenging to correctly determine the disease status for younger generations. To confirm the findings, replication in other datasets or families would be ideal as the identified variants were specific to each family. To further understand the biological effects of these variants and genes in the pathogenicity of MD, in vitro and in vivo experiments are required. 

## 5. Conclusions

In conclusion, we identified seven genetic variants that segregate with familial MD. Four of these variants are located in genes linked to the same co-expression network with previous familial MD candidate genes. Our findings support the role of oxidative stress in the pathology of familial MD but also suggest dysregulation of mitotic spindle formation, highlighting the genetic complexity behind MD.

## Figures and Tables

**Figure 1 genes-13-00998-f001:**
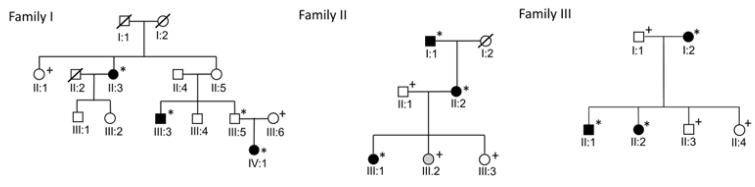
Pedigrees of the families. Individuals with definite MD are marked with black symbols and individual with possible MD marked with gray symbols. Individuals marked with asterisks were included in the exome sequencing and individuals with plus symbols were genotyped.

**Figure 2 genes-13-00998-f002:**
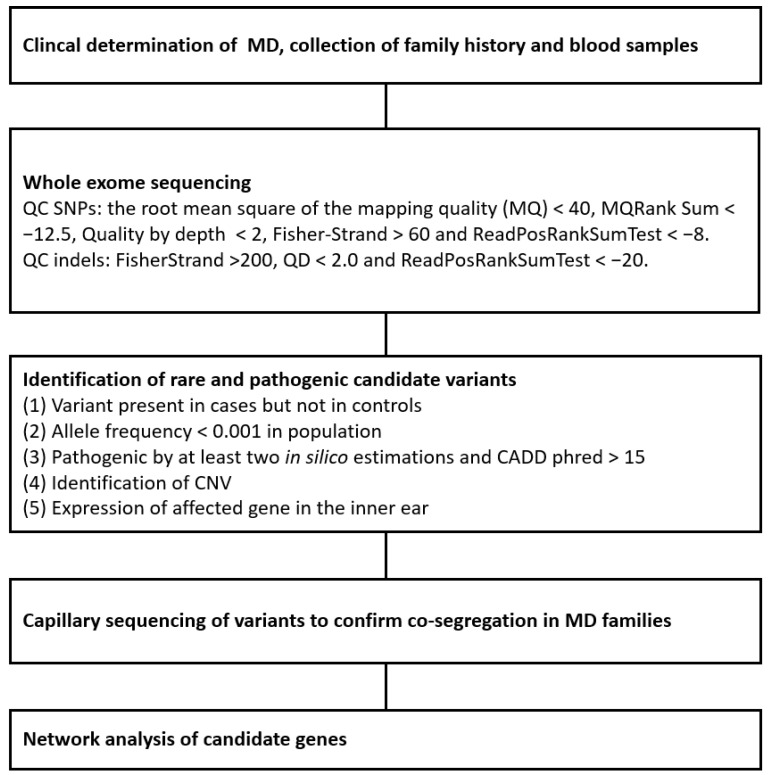
Flowchart of the methods used in this study.

**Figure 3 genes-13-00998-f003:**
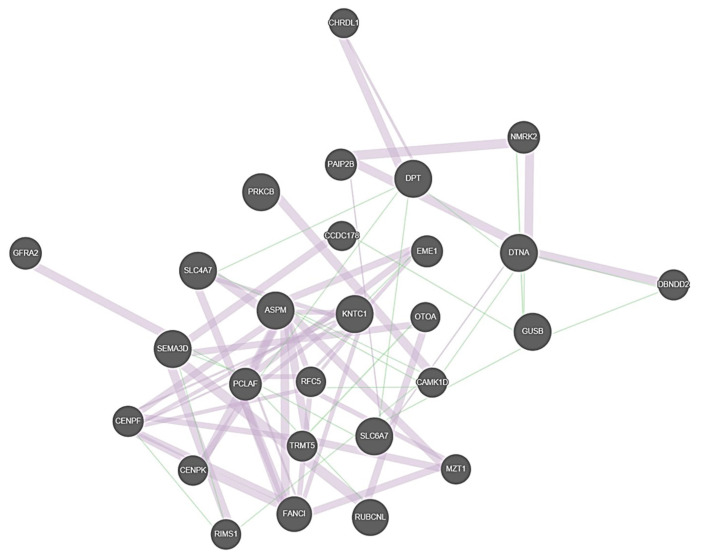
Illustration of the co-expression network between the familial MD candidate genes. Thick purple lines represent co-expression and thin green lines genetic interactions. Genes that were provided as input in GeneMANIA software are marked with cross-hatched circles of a uniform size. Genes added as relevant interactors by the software are shown with solid circles whose size is proportional to the number of interactions they have.

**Table 1 genes-13-00998-t001:** Variants segregating with Ménière’s disease in families I–III.

Family	Gene	Variant	rsID	Type	gnomAD Finn	gnomAD Total
I	*CYP2B6*	c.200C>T; p.T67M	rs138264188	Missense	0.0001194	0.001253
II	*GUSB*	c.323C>T; p.P108L	rs1268678201	Missense	0.0001995	0.00003199
II	*EPB42*	c.1089G>A; p.W363X	rs201351228	Nonsense	0.0009161	0.00008134
III	*OVCH1*	c.1316T>A; p.L439X	rs775251987	Nonsense	0.0006504	0.00008044
III	*KNTC1*	c.5242A>C; p.T1748P	rs1272541364	Missense	0.0001872	0.00001635
III	*ASPM*	c.5207A>G; p.Q1736R	NA	Missense	NA	NA
III	*SLC6A7*	c.322G>C; p.V108L	rs775035174	Missense	0.001314	0.0001238

**Table 2 genes-13-00998-t002:** Genotypes of the identified variants. Variants co-segregating with FMD are bolded. Variants for which co-segregation was not established are marked with a light gray color.

Family I	II:1	II:3 ^M^	III:3 ^M^	III:5 ^C^	III:6	IV:1 ^M^
***CYP2B6* c.200C>T**	C/C	C/T	C/T	C/T	C/C	C/T
**Family II**	**I:1 ^M^**	**II:1**	**II:2 ^M^**	**III:1 ^M^**	**III:2 ^P^**	**III:3**
***GUSB* c.323C>T**	C/T	C/C	C/T	C/T	C/T	C/C
* SLC8A2 * c.1792G>T	G/T	G/G	G/T	G/T	NA	G/T
* OTOA * c.1387G>T	G/T	G/T	G/T	G/T	NA	G/T
* FLJ44635 * c. 244C>A	A/A	C/C	C/A	C/A	C/A	C/A
* PNPLA2 * c. 493G>A	G/A	G/G	G/A	G/A	G/A	G/A
* SEC24C * c.665G>A	G/A	G/G	G/A	G/A	G/G	G/A
***EPB42* c.1089G>A**	A/A	G/G	G/A	G/A	G/A	G/G
* AR * c.2395C>G	G/G	C/C	G/C	G/C	G/C	G/C
**Family III**	**I:1**	**I:2 ^M^**	**II:1 ^M^**	**II:2 ^M^**	**II:3**	**II:4**
***OVCH1* c.1316T>A**	T/T	T/A	T/A	T/A	T/T	T/T
***KNTC1* c.5242A>C**	A/A	A/C	A/C	A/C	A/A	A/A
* RYR2 * c.2807C>T	C/C	C/T	C/T	C/T	C/T	C/T
***ASPM* c.5207A>G**	A/A	A/G	A/G	A/G	A/A	A/A
***SLC6A7* c.322G>C**	G/G	G/C	G/C	G/C	G/G	G/G
* ATR * c.6967A>C	C/C	A/C	A/C	A/C	A/C	C/C

M = definite MD, C = known carrier, P = possible MD.

## Data Availability

All the data supporting the results of this study are included in the article.

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
