# Peer review of "New Genetic Variants in CYP2B6 and SLC6A Support the Role of Oxidative Stress in Familial Ménière’s Disease"

_genes, 2022, doi:10.3390/genes13060998_

Round 1

Reviewer 1 Report

This paper described new genetic variants in CYP2B6 and SLC6A in familial Ménière’s disease. Oveall this paper is well-written and interesting to the readers.

But it contains some short comings that should be described. 

  1. As the authors described, familial MD has been observed to have decreased penetrance. Therefore, MD is unlikely to be a monogenic disorder. This paper might mislead the readers to understand MD as a difinite monogenic disorder. 
  2. It is hard to conclude that these variants caused MD because they showed only three pedigrees with a variant in different genes. 
  3. The authors should describe the mechanism of low penetorance especially in family1.
  4. The authors should describe their IRB approval number and its date.

Author Response

Reviewer 1

1. As the authors described, familial MD has been observed to have decreased penetrance. Therefore, MD is unlikely to be a monogenic disorder. This paper might mislead the readers to understand MD as a difinite monogenic disorder.

Response: Reviewer is correct that we mention that familial MD appears to often follow dominant inheritance pattern with incomplete penetrance. However, we also describe the multiple genes and pathways involved. In the introduction we have presented previously reported familial MD studies and say that this suggests multifactorial genetic etiology (lines 47-57). Overall, we agree with reviewers view of the complexity of MD genetics. We have now added text to further emphasized this in the introduction lines 68-69.

2. It is hard to conclude that these variants caused MD because they showed only three pedigrees with a variant in different genes. 

Response: We agree that our study does not provide sufficient evidence to show that these variants would be causal and we have not stated this in the manuscript. Rather we consider that these variants may predispose to MD in these families. We consider MD to have a multifactorial background and identification of predisposing genetic variants and candidate genes provides better understanding of the pathological pathways in MD. Other studies have also reported that predisposing variants specific to individual families (1-3). We have performed a network analysis to show that four of the here reported candidate genes can be linked with previously identified candidate genes, suggesting that although individual variants may be family-specific, same molecular pathways are affected. We have now added text to further clarify our conclusions in the manuscript page 6, lines 243

3. The authors should describe the mechanism of low penetorance especially in family1.

Response: To understand the mechanism leading to low penetrance in these families, experimental studies would need to be conducted to further investigate the biological role of each variant and gene. Even still we would not completely be able to elucidate the mechanisms for low penetrance as this is likely to be affected by several biological and clinical factors as well as interaction of genetic and environmental components. We feel that that is a subject for a separate study.

4. The authors should describe their IRB approval number and its date.

Response: I assume that IRB refers to institutional review board. In Finland, where the study was conducted, the practice is that the ethics committee of the local hospital district will evaluate the study. This study was approved by the Ethics Committee of the Northern Ostrobothnia Hospital District 20.12.2000 (Approval number 131/2000). All patients were informed of the risks involved in obtaining a venous blood sample. A written consent was obtained from all participants. We have added text regarding ethics in lines 87-89.

References:

  1. Martin-Sierra, C., et al., Variable expressivity and genetic heterogeneity involving DPT and SEMA3D genes in autosomal dominant familial Meniere's disease. Eur J Hum Genet, 2017. 25(2): p. 200-207.
  2. Martin-Sierra, C., et al., A novel missense variant in PRKCB segregates low-frequency hearing loss in an autosomal dominant family with Meniere's disease. Hum Mol Genet, 2016. 25(16): p. 3407-3415.
  3. Requena, T., et al., Identification of two novel mutations in FAM136A and DTNA genes in autosomal-dominant familial Meniere's disease. Hum Mol Genet, 2015. 24(4): p. 1119-26.

Reviewer 2 Report

Overall, the study idea is interesting, and the work can have merit in the related field if all the validation process have been done. The following concerns should be addressed

General comment

The authors should ensure that gene names are italicized to match the standards of HUGO for gene nomenclatures in the “Title” and “abstract section” as they did throughout the manuscript.

Methods

- Line 76: “AAO-HNS” should be written at its first mention by the long name and a reference should be provided to support the information.

- Line 81:” 76 unrelated individuals” did they have the same ethnicity as cases?

- As part of data transparency, the authors should elaborate more on “using standard protocols”, what are these protocols and please cite a reference to support your data.

Whole exome sequencing

- Did the authors measure their extracted DNA before being subjected to sequencing?

- It is highly recommended that authors provide a flowchart for the work and the filtering process to clarify the stages and steps of the work, including the wet lab and the in silico analyses to facilitate following the authors in their elaboration.

- The work validation by Sanger sequencing for the DNA (at least for the probands) should follow the WES to validate the identified potential variants.

- The complete long names of the applied tools should be clarified at their first mention in this section.

- Lines 107 and 109: the authors should revise the way of reference citation to keep consistency throughout the manuscript.

- Lines 115 and 116:” Variant was considered pathogenic when it had CADDpherd value > 15 and it was considered pathogenic by at least two of the LJB database estimations” this needs more elaboration and clarification for the readers  …. On what basis this cut off was selected? What is the algorism on which this tool work and differentiate pathogenic vs. non-pathogenic variants?

 Results

- The general characteristics (i.e. the basic demographics data, including the history of consanguinity, and the related clinical data: age of diagnosis, duration of disease, …etc.) of the probands were not clear. Also, did any associated morbidities are present? What about the treatment history?

- Table 2 footer: The meaning of black/gray text should be clarified.

- Figure 2 legend: What did the size of each dot indicate? Should also be clarified in the legend.

- Some examples of the visualized WES reads for the specified identified variants should be provided to support the results.

- A summary report of the WES data analysis should be provided to support section 2.3.

-The results of the functional analyses and the functional impact of the identified variants were not clear. For example, the authors run several tools to test the pathogenicity of the identified variants as (SIFT, PolyPhen2 113 HDIV, PolyPhen2 HVAR, LRT, MutationTaster, MutationAssessor, FATHMM, GERP++, 114 PhyloP and SiPhy). What were the results of these analyses for the segregated seven variants identified in this work?

Discussion

- The authors should provide the study limitation(s) by the end of the discussion.

Minor comment

- Line 53: the long name of DPT was not clear.

- After each website addressed in the manuscript, the authors should write the last date of accession (day, month, year).

- Line 239: please revise “we seven identified genetic variants”

Author Response

1. The authors should ensure that gene names are italicized to match the standards of HUGO for gene nomenclatures in the “Title” and “abstract section” as they did throughout the manuscript.

Response: We have now checked that all gene names are italicized to match the standards of HUGO for gene nomenclatures.

2. Methods

a) Line 76: “AAO-HNS” should be written at its first mention by the long name and a reference should be provided to support the information.

Response: We have modified the text accordingly and added reference to the manuscript.

b) Line 81:” 76 unrelated individuals” did they have the same ethnicity as cases?

Response: Yes, all samples have same ethnicity and are from the same geographical region in Finland. We have clarified this in the text.

c) As part of data transparency, the authors should elaborate more on “using standard protocols”, what are these protocols and please cite a reference to support your data.

Response: Here “using standard protocols” referred to DNA extraction from blood samples on page 3. We have now specified the extraction method used.

3. Whole exome sequencing

a) Did the authors measure their extracted DNA before being subjected to sequencing?

Response: Yes, the quality of DNA was measured using nano drop to determine concentration and purity of samples. The commercial laboratory at BGI Hongkong also measured the DNA using Qubit before library construction. We have clarified this in the text on page 3 lines 100-102.

b) It is highly recommended that authors provide a flowchart for the work and the filtering process to clarify the stages and steps of the work, including the wet lab and the in silico analyses to facilitate following the authors in their elaboration.

Response: We have added flowchart as Figure 2.

c) The work validation by Sanger sequencing for the DNA (at least for the probands) should follow the WES to validate the identified potential variants.

Response: We have validated the WES findings using capillary sequencing while also using capillary sequencing to genotype additional family members. This is mentioned in line 135-137 in the methods section.

d) The complete long names of the applied tools should be clarified at their first mention in this section.

Response: We have added long names of tools when possible. Not all tools have long names or the abbreviation has become their established name.

e) Lines 107 and 109: the authors should revise the way of reference citation to keep consistency throughout the manuscript.

Response: Unfortunately, we are not sure what references this considers as there are no references on the lines 107-109. We have checked that all references in the manuscript follow the appropriate style.

f) Lines 115 and 116:” Variant was considered pathogenic when it had CADDpherd value > 15 and it was considered pathogenic by at least two of the LJB database estimations” this needs more elaboration and clarification for the readers  …. On what basis this cut off was selected? What is the algorism on which this tool work and differentiate pathogenic vs. non-pathogenic variants?

Response: We used ANNOVAR to retrieve pathogenicity estimations for variants using CADDpherd and LJB database. LJB database also kown as dbNSFP has collection of pathogenicity estimations retrieved from several algorithms (SIFT, PolyPhen2 HDIV, PolyPhen2 HVAR, LRT, MutationTaster, MutationAssessor, FATHMM, GERP++, PhyloP and SiPhy). For these we assumed that for any variant to be pathogenic it would have to be estimated such by at least two separate algorithms. Although the mathematical background for these algorithms differs they generally use for example information of locus conservation (the more conserved the amino acid is the more likely it’s mutations is to be harmful as there is likely to be selective pressure working against introduction of variation) and amino acid properties such as polarity, charge and size etc (the more the new amino acid differs from the original, the more likely it is to alter protein function). The Combined Annotation Dependent Depletion i.e. CADD scores the predicted deleteriousness of single nucleotide variants and insertion/deletions variants in the human genome by integrating multiple annotations including conservation and functional information into one metric. Variants at the top10% of CADD scores have a phred score of 10 and top 1% have a CADD phred score of 20. The typical cut off for CADD phred score is 10-25, we used cut off of 15. This approach of using together several pathogenicity estimations allows us to identify variants that may alter the protein function with high confidence.

Results

a) the general characteristics (i.e. the basic demographics data, including the history of consanguinity, and the related clinical data: age of diagnosis, duration of disease, …etc.) of the probands were not clear. Also, did any associated morbidities are present? What about the treatment history?

Response: The following information of the families have been described in previous publications (1,2). We have clarified this in the text.

  1. Hietikko E, Kotimäki J, Kentala E, Klockars T, Sorri M, Männikkö M. (2011) Finnish familial Meniere disease is not linked to chromosome 12p12.3, and anticipation and cosegregation with migraine are not common findings. Genet Med. 13(5):415-20
  2. Hietikko E, Kotimäki J, Sorri M, Männikkö M. (2013) High incidence of Meniere-like symptoms in relatives of Meniere patients in the areas of Oulu University Hospital and Kainuu Central Hospital in Finland. Eur J Med Genet. 56(6):279-85.

b) Table 2 footer: The meaning of black/gray text should be clarified.

Response: We have clarified this in the table legend.

c) Figure 2 legend: What did the size of each dot indicate? Should also be clarified in the legend.

Response: Genes that are provided as input in GeneMANIA software are marked with cross-hatched circles of a uniform size, while those that were added as relevant genes by GeneMANIA are shown with solid circles whose size is proportional to the number of interactions they have. We have added this information to Figure 2 legend.

d) Some examples of the visualized WES reads for the specified identified variants should be provided to support the results.

Response: We have added Supplementary Figures 1-7 to visualize locus of each variant in one affected carrier and one healthy family member. This is mentioned in the manuscript page 4 lines 146-147.

e) A summary report of the WES data analysis should be provided to support section 2.3.

Response: We have added a Supplementary Table 1 summarizing the WES data and Supplementary Table 2 summarizing the WES alignment.

f) The results of the functional analyses and the functional impact of the identified variants were not clear. For example, the authors run several tools to test the pathogenicity of the identified variants as (SIFT, PolyPhen2 113 HDIV, PolyPhen2 HVAR, LRT, MutationTaster, MutationAssessor, FATHMM, GERP++, 114 PhyloP and SiPhy). What were the results of these analyses for the segregated seven variants identified in this work?

Response: We have added a Supplementary Tables 3 and 4 showing the detailed quality and annotation results for each of the identified variant.

Discussion

a) The authors should provide the study limitation(s) by the end of the discussion.

Response: We have added text in discussion concerning the study limitations as suggested by the reviewer.

Minor comment

a) Line 53: the long name of DPT was not clear.

Response: We have clarified this in the text.

b) After each website addressed in the manuscript, the authors should write the last date of accession (day, month, year).

Response: We have revised this according to reviewers’ suggestion

c) Line 239: please revise “we seven identified genetic variants”

Response: We have revised this according to reviewers’ suggestion

Reviewer 3 Report

The only remarks concernant the paper are the following:

  1. In line 76 instead "Three MD Families with..." I think that is better "Three MD Families each with..."
  2. In table 2 the significance of the terms write in light gray must be presented in the legend
  3. In figure 2 the discrimination between the colour green and purple of lines is practically impossible. I consider that is necessary to change the colour or to grow up the dimension of lines.
  4. Line 239  instead "In conclusion, we seven identified..." I think that is better "In conclusion, we identified seven ..."

Author Response

In line 76 instead "Three MD Families with..." I think that is better "Three MD Families each with..."

Response: We have revised this according to reviewers’ suggestion

In table 2 the significance of the terms write in light gray must be presented in the legend

Response: We have revised this according to reviewers’ suggestion

In figure 2 the discrimination between the colour green and purple of lines is practically impossible. I consider that is necessary to change the colour or to grow up the dimension of lines.

Response: We have added clarification to the Figure legend that not only the color but also the thickness of the line visualizes the different interactions.

Line 239  instead "In conclusion, we seven identified..." I think that is better "In conclusion, we identified seven ..."

Response: We have revised this according to reviewers’ suggestion

Round 2

Reviewer 2 Report

The authors have adequately addressed the concerns raised by the reviewer. Thank you